# Shirley Caesar and the Politics of Validating Sexual Agency

Angela Marie Nelson [ID]

School of Cultural and Critical Studies, Bowling Green State University, Bowling Green, OH 43403, USA;
anelson@bgsu.edu

**Abstract:** Black gospelwoman and pastor Shirley Ann Caesar Williams, better known as "Shirley Caesar" to her listeners over the last several decades, entered a professionalization phase of her ministry and career from 1958 to 1966 when she joined and performed with The Caravans. Being a Caravan member brought with it the possibility of being in compromising situations while on the road, which would expose Caesar—a Black woman—to the possibility of sexual (and racial) violence. Caesar's first night as a Caravan in 1958 provided such a circumstance, a failed sexual advance that she describes in Chapter 5, "On the Road with the Caravans", of her 1998 autobiography, *The Lady, the Melody, and the Word*. Caesar's identities (Black, woman, Christian, chaste) intersecting with a potential sexual advance and her reaction to it is fodder for the reinforcement of Black male authority, power, privilege, and dominance in the Black Sanctified Church as well as the assertion of sexual agency. Today Caesar continues to shape her complex public identity born out of a set of negotiations embracing and challenging specific gendered, racial, sexual, and religious norms, the conditions of Black and white mobility, and patterns of religious authority. However, for her, religious authority remains paramount.

**Keywords:** Shirley Caesar; sexual purity; Black Sanctified Church; sexual agency

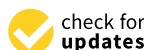



## 1. Introduction

Black gospelwoman and pastor Shirley Ann Caesar Williams, who has been singing gospel music for nearly 70 years, began doing so at eight years old—"I was the little girl with the enormous voice" (Caesar 1998, pp. 3, 36). Better known as "Shirley Caesar" to her listeners over the last several decades, Caesar sang in organized musical groups at a young age. These groups included ones composed of her family members and non-family members—local groups such as the Charity Singers of Durham and Thelma Bumpass and the Royalettes (Caesar 1998, pp. 36–37). Caesar's gift for singing brought her to the attention of the popular Black women's gospel group The Caravans, the "best-known Black female gospel group of all time". Caesar joined and performed with The Caravans in August 1958, just a few months before her twentieth birthday (Caesar 1998, pp. 61–62, 68). From 1958 to 1966, she began making a living as a gospel singer that involved attention to business practices, traveling and touring, wages and labor, wardrobe and makeup, and vocal and stage performance, entering a professionalization phase of her ministry and career.

Caesar touring with The Caravans meant moving to Chicago, Illinois, away from her hometown in Durham, North Carolina. Staying in under-par hotels reserved for African Americans became a normal part of Caesar's life. As a Caravan member and now participating in the mobility of Black gospel music on a national scale, this brought with it the possibility of being in compromising situations while on the road, which would expose Caesar—a Black woman—to the possibility of sexual (and racial) assault and violence. Traveling provided a certain type of "freedom" for Black gospelwomen to spread the gospel of Jesus, but it also brought with it the reality of spatial limitations of race and gender. Carol Brooks Gardner found that in urban settings, as a "part of their roles", children, African Americans, and women are "open to the public", meaning these populations are open to uninvited, unwarranted, or unprovoked public commentary (Gardner 1980, p. 333). This

mobility, being in major cities, being alone, a woman, and an African American, made Caesar an "open person" on at least two levels: one determined by sex and the other determined by race (Gardner 1980, p. 333). Caesar's first night as a Caravan in 1958 provided such a circumstance, a failed sexual advance that she describes in Chapter 5, "On the Road with the Caravans", of her 1998 autobiography, *The Lady, the Melody, and the Word: The Inspirational Story of the First Lady of Gospel*. Caesar's identities (Black, woman, Christian, chaste) intersecting with a potential sexual advance and her insistence on maintaining her sexual purity throughout the potential encounter is fodder for the reinforcement of Black male authority, power, privilege, and dominance in the Black Church but also for the expression of her sexual agency. Caesar's insistence on sexual purity highlights the *de jure* and *de facto* limitations and restrictions placed on Christian Black women's bodies largely sanctioned, maintained, and governed by the beliefs and values of religious Black men as well as the possibility of considering her insistence as a decision Caesar makes to align herself with the Holy Spirit (Moultrie 2017, p. 24). This decision to align herself with the Holy Spirit signals Caesar's sexual agency. Caesar's sexual agency is her liberty to take any action she chooses regarding sexual activity (or inactivity) with an understanding this action may ultimately be harmful or helpful to her spirit, psyche, or body (Moultrie 2017, p. 12).

## 2. The Specter of Holy Consensual Sexual Seduction

Traveling and touring with The Caravans took Shirley Caesar away from the familiarity of her family, church, and local–regional "gospel network" (Caesar 1998, pp. 37–38). Caesar was born on 13 October, 1938, to James "Big Jim" Caesar and Hallie Martin Caesar in their home on 2209 Chatauqua Street, the tenth of thirteen children (Caesar 1998, p. 33; Caesar n.d.). Caesar's father led a male gospel group, the Just Come Four Quartet, and worked for Liggett and Myers Tobacco Company in Durham (Caesar 1998, pp. 13–14). Caesar's mother also worked at Liggett and Myers for 13 years as a tobacco stemmer. Changes in the industry as well as health issues compelled Hallie Caesar to leave Liggett in 1939 (Harold 2020, p. 71). Caesar singing and traveling at an early age is partly due to her mother's condition coupled with the death of her father when she was eight years old (Caesar 1998, pp. 18–20). Caesar's formative religious upbringing was in two Black sanctified churches,[1] Fisher Memorial United Holy Church of America and Mount Calvary Holy Church (Caesar 1998, p. 23). Black sanctified churches rose from the Holiness-Pentecostal movements of the late nineteenth and early twentieth centuries, emphasizing the experience of baptism in the Holy Spirit and living a life of holiness. These churches, in opposition to the Black Baptists and Methodists who assimilated and imitated the cultural and organizational models of European-American patriarchy, were set apart by virtue of their codes of morality and their peculiar liturgies of song, speech, and dance that included African-derived oral music traditions, ecstatic praise traditions, and charismatic preaching and singing (Gilkes 1985, p. 77; Sanders 1996, p. 143). Almost as long as Caesar has been a singer, she has been a preacher. Caesar notes that her calling to preach occurred when she was 17 years old (Caesar 1998, pp. 10, 29–31, 84, 174, 178, 184). Indeed, part of the reason Caesar left The Caravans was because it was difficult to fulfill her responsibilities to the group and to God (Caesar 1998, p. 84). "Our rigorous, inflexible concert itinerary made it almost impossible for me to schedule any speaking engagements and, to a large degree, deprived me of the liberty to accept the invitations I frequently received to conduct revival services" (Caesar 1998, p. 84). Caesar continues to preach and to sing to this day.

Caesar's family and church inculcated "Christian beliefs" in her: "Of all the factors that held our neighborhood and our lives together, none was more influential than the church. From the very beginning, I was immersed in the gospel tradition, in both my family and church. We prayed together, worshiped together, and lived out our Christian beliefs in our daily lives" (Caesar 1998, p. 23). The church taught Caesar the "basics" of her faith and the "truths" she learned that "became the foundation" of her life and her music (Caesar 1998, p. 25). The fact that Caesar was a young, single Black woman moving in and outside

of the Black Sanctified Church, and around the U.S., was significant. Traditionally, the social place of women was in private spaces such as the home. Being in public spaces was the domain of men. Caesar's singing voice, her musical vocality, is what permitted her entry into the diverse public spaces that she did. Caesar's family and church provided spiritual protection while in those new spaces. "With the armor of God and the protective covering of my mother's prayers, I survived attacks like that on the road" (Caesar 1998, p. 68). As she would detail later in her autobiography, The Caravans became Caesar's new community "on the road". Caesar went home as often as possible when The Caravans did not have any engagements. In the meantime, The Caravans provided the safety and protection Caesar recognized to be a help to her. "They would often tell me, 'Because you are new on the road, you must be watchful'" (Caesar 1998, p. 79). The word "watchful" (as Caesar and The Caravans no doubt knew very well) is spoken by Jesus and referred to in the writings of Paul in the New Testament of the Bible (The Holy Bible 2016b, Col. 4.2). When The Caravans counseled Caesar to be "watchful", they were warning her to be prayerfully aggressive concerning and spiritually alert to temptations and being taken advantage of.

Caesar spent her first day as a Caravan traveling to meet them in Washington, D.C. The Caravans were performing in Chicago and they planned to meet Caesar at the Casbah Hotel in D.C. after traveling from Illinois.[2] The Casbah Hotel was located on U Street, a historically significant area for African Americans encompassing businesses, hotels, schools, residences, and theaters. Caesar traveled there by bus and notes that it was her first time staying in a hotel (Caesar 1998, p. 67). "I didn't feel safe there, and it wasn't much of a welcome for a young girl away from home by herself for the first time. I felt so alone" (Caesar 1998, p. 67). Caesar's "new world" of traveling with The Caravans involved a Black male gospel singer whom Caesar met in the hotel lobby (Caesar 1998, p. 67). The group of which he was a part was also staying in the Casbah Hotel. The group was to perform in the same concert as The Caravans the next evening (Caesar 1998, p. 67). Caesar had eaten after arriving at the hotel and decided to go to bed early "hoping and praying that by morning Albertina [Walker, leader of The Caravans] and the other group members would be there" (Caesar 1998, p. 67). "With great fear" in her heart, Caesar "locked the door with the chain and dead bolt" (Caesar 1998, p. 67).

Not long after returning to her room, Caesar heard a knock on her door followed by a man's voice. Caesar went to the door, kept the chain attached, unlocked the door, and looked out. She saw the male gospel singer she had met earlier in the hotel lobby standing there with a "silly grin on his face" (Caesar 1998, p. 67). After asking him what he wanted, this unnamed singing gospelman said, "'I was wondering if I could come in and get you to bless my cross for me?' he said. He held out this huge cross he was wearing around his neck" (Caesar 1998, p. 67). In response, Caesar "slammed the door and shouted through the wall, 'I'll do it tomorrow.' And that was the end of that" (Caesar 1998, p. 67). Unbeknownst to Caesar, her first night as a Caravan had held the specter of "holy consensual sexual seduction"—a process by which Caesar could have consented to a sexual encounter with the singing gospelman out of respect for the potential reward of holiness and sanctification because of the encounter.

The request for a "blessing" was not unusual or unknown to Caesar. A "blessing" is a declarative prayer one person prays on behalf of another person. This prayer serves as a form of approval for the recipient who is usually in a subordinate position in relation to the person who prays for them. Caesar knew the purpose of a blessing. For example, Caesar sought her mother's approval, or blessing, before consenting to join The Caravans. "After praying [Mama] now felt that this was God's will for my life. She said to me, 'All you have ever wanted to do is to sing for the Lord. Go with my blessings'" (Caesar 1998, p. 66).

This nameless singing gospelman's decision to single out Caesar specifically to "bless" his cross and to make this request of her in a hotel room rather than at a church altar, for example, is problematic. In fact, several aspects of his actions are suspect. For one, why did he go to Caesar's hotel room in the evening when he could have asked Caesar to bless his cross when they were in the hotel lobby earlier? Two, he was a part of a male gospel group.

Could not they have blessed his cross? Three, why did he not ask his church, including the pastor or its members, to bless his cross before he went to Washington, D.C., with the group in the first place?

The male gospel singer chose Caesar because she was a young woman on the road alone. He was relying on the common subordinate positioning of the woman in the Black Sanctified Church (Butler 2007, pp. 52–53). This subordination is not concerning the gifts of the Spirit such as singing, praying, and prophesying but rather concerning ruling and governing authority. In other words, men are the preferred sex to be in roles such as bishops, pastors, and elders who lead or govern their churches (Gilkes 1985, pp. 682–83). Women, while leading women's groups, organizing meals for special church gatherings, and supervising children, largely support the men as ruling leaders (Stephenson 2011, p. 413). The male gospel singer felt no hesitation about approaching Caesar since "he had used that same line to lure and seduce . . . many young women" before (Caesar 1998, p. 68). He attempted to and engaged young Black women victims in holy consensual sexual seduction. By requesting specifically a "blessing", he moved his motives from being seemingly physical to assuredly spiritual. One can speculate that these young Black women such as Caesar consented to being seduced because they believed that the resultant sexual activity served a holy purpose and was complying with a biblical doctrine. Biblically grounded, as in relation to a "blessing", they consented. After all, prayers of blessing are legitimate, so it stands to reason that other activities surrounding it are legitimate as well. One can speculate that the unnamed male singer reasoned that since Caesar, a Christian woman, was committed to God, she would positively respond to his request. If she responded positively to the male singer's request, then surely, led by her Christian beliefs and values and normalized deference to men, Caesar would allow him to enter her hotel room. Entrance into Caesar's hotel room was important because after entering her hotel room, the male petitioner, by now sanctioned through her consent to the "blessing", could go further in his pursuit of physical intimacy and sexual intercourse.

## 3. Black Gospelwoman's Purity

The inclusion of this story—of a failed sexual advance—reveals Caesar's own firmly held values about her sexual purity as a young sanctified Black woman—the chain she kept on her door represents her beliefs clearly. Sexual purity in Black sanctified churches is defined as avoiding sexual intimacy outside of heterosexual marriage. However, as Monique Moultrie notes, the meaning behind sexual purity varies based on race and context (Moultrie 2017, p. 24). Caesar being on the road exposed her to the possibility of sexually motivated "attacks", even if deviously couched in "holy" terms, yet also strengthened her commitment to her personal purity, chastity, and desire to maintain a holy and sanctified body. This is important because any kind of sexual impropriety— masturbation, premarital sex, adultery, same-sex desire and acts—would conflict with Caesar's ethics of accountability to God, or her submission to "God's gaze" upon her life (Moultrie 2017, p. 23; Frederick 2016, p. 92). "God's gaze", no doubt, prompted Caesar to exercise her sexual agency.

Holiness and sanctification are important values in the Black Sanctified Church, of which Caesar was and is a member. Marla F. Frederick describes the early Pentecostal expectations for personal purity as "strikingly rigid and influential" (Frederick 2016, p. 91). Indeed, traveling gospelwomen and their sanctified faith community held entrenched values and ideals of sexual purity and restraint that highlight the significance of Black gospelwomen's intersectional identities of gender, sex, race, sexuality, and religion.[3] In short, African American sanctified churchwomen (and gospelwomen) bore the weight of gender-specific sexual expectations because of Christian denominational doctrine and practice, white racism, and "masculinist prescriptions of purity" (Frederick 2016, p. 89). While chastity and complete celibacy before marriage are required for sanctified living, celibacy assures women of the sanctified church, including Caesar, and the men they marry that the women are wholesome and untouched by other men. Monique Moultrie notes that

"pure living" gave Black sanctified churchwomen (and gospelwomen) "more credibility as [they] continued teaching and preaching" (Moultrie 2017, p. 25). Caesar herself continues to express a strong conviction about values of chastity. "Save yourself sexually for the mate God has for you" (Caesar 1998, p. 135).

Regarding the Casbah Hotel incident, Caesar describes herself as a "young, innocent girl from a small town in North Carolina". "I came to the Caravans a very naïve young girl, straight out of bobby sox . . . and extremely unaware" (Caesar 1998, p. 79). Her self-identification as "young", "naïve", "innocent", and from a "small town" in the South implies virginal purity. Indeed, Caesar's early professional traveling stage, or pulpit, name was "Baby Shirley". "I was small in stature, very petite, and wore my hair in braids. When I sang they had to stand me either on a box or a table" (Caesar 1998, p. 3). The use of "baby" directs attention to her age and with it the assumption of youthfulness, naïveté, and innocence. In other words, possessing sexuality as a "baby" is impossible. Naming herself "Baby" reinforces Caesar's separation from and need for support and protection from the adults around her. Being named "Baby Shirley Caesar" also reminded the men preachers with whom she traveled and ministered that she was unavailable to them sexually. Projecting a child persona (because she in fact began singing within her intimate family and church circles when a child) emphasizes Caesar's innocence, ignorance, and chastity. However, although young and innocent, Caesar proved to be *not* naïve. She was certain she knew what her fellow singing gospelman really wanted: "I wasn't stupid. I wasn't about to fall for that kind of line" (Caesar 1998, p. 67).

This child persona and actual young age perhaps caused men preachers to take advantage of her financially. A custom in the Black Sanctified Church is to raise "freewill offerings", (or "love offerings"), to pay a guest preacher and singer for sharing their ministry gifts (Lincoln and Mamiya 1990, p. 146). A component of the Black gospel gift economy, Caesar and her family personally benefited from freewill offerings. "Whenever I received an offering from singing in churches I brought every penny home and put the money in Mama's hand" (Caesar 1998, p. 53). One story she relates in detail involves a Black preacher who did not pay her equitably. Although the story illustrates Caesar's knowledge of how the Black gospel gift economy was to operate, it also emphasizes the gender and age differential that Caesar experienced as "Baby Shirley Caesar" (Caesar 1998, p. 36). While Caesar was well prepared to safeguard her sexual purity, in what follows in a slightly lengthy passage, she was unprepared to safeguard her financial security.

> Early in my singing career, before I sang with Leroy Johnson, I was traveling with a preacher named Reverend Jones. We had traveled for one week together, going from church to church. He preached and I sang. As was the custom, a freewill offering was collected for us at the end of the services.

> At the end of the week when it was time to be paid, I went to Reverend Jones expecting him to give me my share of the offering.

> He smiled and placed two quarters in my hand. He patted me on the head and told me to go buy myself an ice-cream cone. I looked down at those two quarters and thought within myself, I can't take only fifty cents home to Mama. Reverend Jones said, "The Lord is going to bless you".

> Even though I knew the Lord would surely bless me, before I realized it I had blurted out, "My blessing is in your pocket".

> He laughed a bit nervously, then pushed my hand away and said, "Don't sass me, girl".

> I never received more than fifty cents from him that day, and I didn't like it when I had to go home with such a small amount to give Mama. (Caesar 1998, pp. 40–41)

Here Caesar was a young girl partnering with an adult male. Reverend Jones reminded Caesar of her place as a youth in relation to him when he gave her fifty cents for her share of the freewill offering from that week, when he patted her on the head, and when he told her to buy an ice-cream cone. Caesar notes that freewill offerings could range from $100 to $300

(Caesar 2018). Indeed, comparatively, fifty cents was a "small amount". Specific gender and age-related overtones reside in Reverend Jones' response to Caesar's objection to some future blessing she would receive from the Lord: "Don't sass me, girl". The gender and age differential required a certain level of respect and trust from Caesar towards Reverend Jones. First, Caesar, an adolescent girl, had to show respect to Reverend Jones because he was an adult, and second, she had to show respect to Reverend Jones because he was a preacher. In Caesar's faith community, being called a preacher implies that one is male. Implicit in this religious communal norm, to respect Reverend Jones because he was a preacher was to respect him because he was a man.

In the end, Caesar received her "share of the offering" through an informal, unofficial, and unethical barter exchange. Caesar found retribution for the wrong done to her by taking Reverend Jones's pears.

> But I have to confess, one Halloween night shortly after that incident, my friends and I sneaked up on Reverend Jones's front porch and took the chairs he had sitting out there. We used those chairs to climb his pear tree and take all of his pears. I took those pears home, and Mama, not knowing where I had gotten them, made pear preserves and pear pies. Reverend Jones never paid me my money, but indirectly he helped me take care of Mama, because we certainly enjoyed those preserves and pies. After that, I didn't sing with him [anymore]. (Caesar 1998, pp. 40–41)

In this early experience, Caesar's response to what she saw as an abuse of authority was to choose to no longer travel with the Reverend Jones. However, more is happening here when connecting Reverend Jones' incident to her potential sexual encounter at the Casbah Hotel. Caesar anticipated the possibility of being sexually exploited while on the road. She, on the other hand, did not anticipate being financially exploited while on the road. The singing gospelman who knocked at her hotel room door by night and the preaching gospelman who ministered alongside her by night were reflections and refractions of the other. Caesar demonstrated her sexual agency (protecting her sexual purity) in dealing with the singing gospelman. We can assume that Caesar exercised her financial agency (or some type of personal financial management and surveillance) up until her experience with Reverend Jones, the preaching gospelman. Since Caesar was young, it is possible that an older family member helped to ensure that she was paid appropriately and accurately. Caesar recalls that as a child, "whenever I traveled my mother always took precautions for my safety ... She insisted that a family member always accompanied whomever I was traveling with" (Caesar 1998, p. 38). Often, that family member was Caesar's Aunt Ida or brother Cleo (Caesar 1998, p. 3).

The freewill offering which Caesar became accustomed to receiving after she sang in a church was a form of financial agency among the members of Black sanctified churches. It was a way to compensate or to express appreciation for someone's ministry gifts. The freewill offering, connected to the Black gospel gift economy, was practiced by African Americans as the result of them engaging with their Christian faith and in response to their poverty and experiences of discrimination and segregation in the eighteenth, nineteenth, and early twentieth centuries. One reason Caesar and other African Americans who ministered in word and song were driven to share their spiritual gifts in this manner was because of their limited economic opportunities regarding well-paying jobs. Freewill offerings were an agentive practice because these offerings were established and maintained by and among the Black faith community themselves. Therefore, Caesar being exploited by Reverend Jones was doubly punishing. Reverend Jones was taking advantage of one of his own people. In response to this outright financial exploitation, one can speculate that Caesar learned the importance of her financial agency and the need to protect her financial security.

At the Casbah Hotel, her participating in the mobility of gospel—something Caesar now saw as necessary to her career as a female gospel singer—is partially responsible for creating the conditions in which Caesar found herself. Here, however, it was Caesar's

concern about her sexual purity and restraint that underlies a portion of the fear she felt concerning being alone in the hotel. Caesar's response to the Black singing gospelman who knocked at her hotel room door in 1958 also reveals her anxiety about possibly being marked as sexually promiscuous, or as a jezebel. Jezebel is a sexually aggressive woman (Collins 2000, p. 81; Douglas 1999, pp. 36–40). Caesar did not want her sexuality rendered as the "opposite (and absence) of ladydom" due to "jezebelian" behaviors (Lomax 2018, p. 10). Essential to the "black 'nuclear' project", Tamura Lomax describes the "black lady" as reimagining Black womanhood as having dignity and access to a race, gender, and class rewards system for performances of respectability and hard work (Lomax 2018, p. 125). For Caesar, being a Black lady within the Black Sanctified Church meant access to a "divine", or spiritual, rewards system for performances of sexual purity and restraint.

What Caesar was afraid of happening to her in her hotel room and what she knew or conjectured had happened to other young Black women was sexual seduction, which could lead to non-marital sexual intercourse that in turn could result in pregnancy. Any sexual activity prior to marriage was inappropriate and premature in Caesar's eyes and those of her faith community. "God intended for sex to be shared between two individuals only within the sanctity of marriage. Save yourself for the spouse that God has for you" (Caesar 1998, p. 135). Furthermore, any sexual activity would be heterosexual. "Like any single lady I desired to have the American Dream, a husband, children, and a nice home, but I wanted a mate who would understand and support my call to the ministry" (Caesar 1998, p. 129). Even as Caesar is revealing to the reader her American heteronormative hopes and desires, an awareness of male domination and women's subjugation lingers in the background, as she wanted a mate "who would understand and support" *her* call to ministry.

Frederick asserts that through the neo-Pentecostal messages of Black and white women televangelists of the 1990s to the present, Christian women can name and address their pain caused by male sexual indulgence and sexual abuse (Frederick 2016, p. 129). This salvific and restorative opening for women is profound considering the historical practice of silence and invisibility many Black and white women have experienced within male-dominated narratives and structures. Caesar mentioning this story is profound and yet unsurprising given the dichotomous discourses of visibility/invisibility and loudness/silence regarding Black female sexuality in the Black Sanctified Church. Interdictions of Black female sexual visibility are apparent in the Black Sanctified Church when Black girls are told to keep their legs crossed, to cover their breasts, legs, and hips, and are policed regarding spending time alone with a boy (Butler 2007, pp. 77–86). The reasoning is that their appearance could cause strong and weak men and boys alike to be sexually aroused. Therefore, girls and women must control and diminish themselves, conceal their bodies, and consider carefully the spaces they can traverse in order to shore up and to protect men and boys from their "natural" weaknesses (Butler 2007, p. 80; Lomax 2018, pp. x–xii; Moultrie 2017, p. 26). The cultural stakes were higher for the girls, and therefore, conformity was necessary. The leaders of the Black Sanctified Church today continue constantly to instruct Black Christian women and girls in, and expect them to exercise, sexual purity and restraint (see Frederick 2016; Lomax 2018). At the same time, "Expectations and teachings on 'male purity' are virtually absent in Christian churches and communities" (Frederick 2016, p. 105; Moultrie 2017, p. 26).[4] The invisibility/silence surrounding Black Christian women's sexuality manifests in non-acknowledgment of their sexual desires and indulgences, trauma from sexual abuse, assault, coercion, rape, and treatment as second-class citizens to the preferred male sex. Surrounded by male-dominated narratives and structures, Caesar and other Black gospelwomen developed narratives and structures that to this day insist upon their adherence to doctrinal precepts and norms of sexual purity and the protection of God that set them apart from others. Such insistence in the end reinforces male heteropatriarchal dominance but also demonstrates sexual agency.

### 4. Validating Sexual Purity and (or) Sexual Agency

In the Casbah Hotel incident, Caesar narrates a somewhat veiled testimony of personal struggle and triumph over the *temptations* of sexual impropriety rather than the *actions* of sexual impropriety. She expresses gratitude to God for His delivering her from temptation. "I could have fallen into divers temptations . . . But for the grace of God, I could have become an alcoholic, a drug addict, or an unwed mother . . . The power of the Holy Spirit threw a protective shield of spiritual armor around me and gave me strength to resist temptation" (Caesar 1998, p. 41). Theological understandings of such statements as Caesar rests on the belief that God gives supernatural strength to individuals to turn away from temptations and gives the "way of escape" from temptations (The Holy Bible 2016a, 1 Cor. 10.13). Caesar is also demonstrating her sexual agency.

While Caesar did not "miss the mark" of purity, she, nevertheless, validated the mark of sexual purity as legitimate for women in ways that male ministers and parishioners rarely need to do (Frederick 2016, p. 105). Caesar's conviction of sexual purity and restraint (celibacy and abstinence from sexual intercourse with a man) in her life validates sexual purity as the appropriate state of sexuality for a single Black Christian woman before and after marriage, including widowhood but also her sexual agency. The appropriate state of sexuality after singlehood is heterosexual monogamous sexual intercourse with a man, her husband. This validation holds implications for Black women in the Black Sanctified Church and their relationship to Black men and the ruling authority figures in most Black sanctified churches, who are also almost always Black men. This validation also demonstrates sexual agency for Caesar and for Caesar alone.

Caesar's autobiography illustrates the significance of purity and restraint in her life as she elaborates on her experience with a potential sexual advance at the age of 19, how she felt about it, and how she escaped it with God's help and deliverance. Her sexual purity was at stake, and her sexual agency was revealed. Caesar validated that purity and restraint was a standard important to her because of its possible negative ramifications if not adhered to. Although Caesar passed the test, survived the "attack", and did not miss the mark, by discussing this experience, Caesar validated the fact that the mark of sexual purity and restraint was legitimate and was an appropriate (albeit mandatory) standard continually to seek and to maintain (Frederick 2016, p. 105). Caesar also revealed herself to be a fully embodied sexual agent (Moultrie 2017, p. 147).

Caesar's description of her experience marks a gender-specific requirement that held Caesar as accountable to an authority beyond herself to maintain a respectable status as an unmarried woman (Wilson 2019, p. 45). Caesar uses her story of a failed sexual advance to validate her sexual purity for three purposes. One, to state and affirm her allegiance and accountability to her faith community. Two, to state and affirm her allegiance and accountability to God. Three, to affirm her submission and accountability to Black religious male heteropatriarchal ruling authority. Validating her sexual purity would certify her availability for future marriage, credit her present singing and preaching ministry, and confirm her past moment of receiving the calling of God. A fourth purpose is present: Caesar validated her sexual purity to state and to affirm her allegiance and accountability to herself. As Caesar validated her sexual purity, she likewise validated her sexual agency.

Why would Caesar validate sexual purity in her autobiography? Who is Caesar's primary audience other than herself or God? Her primary external audiences include other Black sanctified churchwomen and Black sanctified churchmen. However, men as pastors are the authoritative and ruling audience here. Her story about surviving the sexual "attack" (rhetoric connecting to the "work of the devil" or "attacks of the enemy") is ultimately to assure the Black religious male heteropatriarchy-dominated hierarchy that she is pure. Her sexual purity is an essential criterion to affirm her place in the ministry. This affirmation is needed because women are seen by the Black Sanctified Church as being out of place and out of role (to use Carol Brooks Gardner's words) when they travel and when they minister in word or song. Narratives such as Caesar's provide evidence of ordinary Black gospelwomen that traditional lines of power and domination continue.

Black sanctified women use their sexual purity like a bargaining chip and as social capital within the group. An implied and explicit criterion, sexual purity promotes high self-esteem, expresses respectability, and encourages self-worthiness. Sexual purity is a sign of submission to God, holiness, and sanctification (Moultrie 2017, pp. 24–25). However, the need for male approval and domination makes a private issue (sexual purity) public. While this is true, Monique Moultrie reminds us that even within this heteropatriarchal context, Black sanctified churchwomen (Caesar included) are sexual agents as well.

Caesar's social identities of Christian, Black, woman, and sexually pure intersect, illustrating the ways in which the Black Sanctified Church, African American culture, and Black popular culture reinforce structures of power and dominance (see Lomax 2018). In this story, Caesar is strategically performing social and cultural modes of "being possessed", an extension of family or community belonging, that functions as a layer of credibility for an unmarried young woman who travels outside of her home and home church and sings gospel before Black Sanctified Church congregations within the state and region (Wilson 2019, p. 45). How do we know that Caesar is successful in validating the mark, the goal, and the achievement of sexual purity? There is no documented emergent change in her language and mindset. She offers no new frameworks. Caesar repeats what her religious community, the Black Sanctified Church, has taught her. However, what Caesar does not state explicitly and what is also occurring is validating her sexual agency.

Caesar's final assessment of the incident is this. "With the armor of God and the protective covering of my mother's prayers, I survived attacks like that on the road. The only good thing about that night was that it began to establish the fact among the groups on the road that I was serious about my commitment to Christ" (Caesar 1998, p. 68). Caesar's description of this failed sexual advance as an "attack" requires an understanding of Christian demonology in relationship to God and Jesus. Satan, who was at first an Archangel of God named Lucifer, is frequently referred to in the Black Sanctified Church. A major figure in Christianity, Satan, also known as the Devil, is cast as the chief antagonist and adversary of God, Jesus, and God's people. The Devil represents the primary spiritual source of a person's problems, difficulties, suffering, and struggles. It is the Devil who "attacks" people, causing them to suffer. These attacks can be physical, spiritual, and emotional. The unnamed singing gospelman's request for Caesar to bless his cross was an attack in Caesar's perspective and experience because Caesar perceived it to be a ploy to tempt her to sin. Caesar safeguarded her sexual purity while at the same time demonstrating her sexual agency.

## 5. Conclusions

In her autobiography, Shirley Caesar resists the "culture of dissemblance" that Black women and religious Black women have historically engaged (Hine 1989, p. 915). She openly narrates sexual discourse as she recounts a failed, sexual advance at the Casbah Hotel during her first night as a Caravan in 1958. Black gospel mobility, though a practice that opened economic opportunities to African Americans, at the same time exposed the gendered and racialized burdens that Christian Black women bore while on the road. Caesar's early life is emblematic of the challenges and opportunities presented to a gospel-woman during the important and formative transition to adulthood in the post-war Black gospel economy. Caesar's discussion of the incident at the Casbah Hotel during her first night as a Caravan member shows how Caesar (at first unintentionally), opened a space for sexual discourse (see Lee 2010, p. 93). Although sexual purity is an important sanctified lifestyle imperative, it is not a common topic of gospel song lyrics of which Caesar has sung hundreds of lyrics over the decades. The story (later told in 1998) opened an intended space of discourse by simply being in her autobiography. In so doing, Shirley Caesar was and remains not only a trailblazer in the world of gospel music but is also a trailblazer in including this private and important incident in her autobiography. In her autobiography, Caesar provides one view of Black women's struggles to form positive self-definitions in the face of derogated images of Black womanhood (Collins 2000, p. 102). Today Caesar

continues to shape her complex public identity born out of a set of negotiations embracing and challenging specific gendered, racial, sexual, and religious norms, the conditions of Black and white mobility, and patterns of religious authority. However, for her, religious authority remains paramount.

**Funding:** This research received no external funding.

**Acknowledgments:** I sincerely thank Timothy Messer-Kruse, Jeremy Wallach, and Rebecca Kinney for their feedback on very early drafts of papers about Shirley Caesar. I thank Nancy Spencer for our biweekly meetings where we have discussed our research projects including this project on Shirley Caesar as well as our teaching, research, service, and life. I thank the three anonymous readers who pushed me to go deeper, which improved this paper. Finally, I thank Katerina Rüedi Ray for her guidance, mentorship, and feedback on numerous versions of this paper and others about Shirley Caesar.

**Conflicts of Interest:** The author declares no conflict of interest.

## Notes

1　"In the late nineteenth and early twentieth centuries, beginning at the end of Reconstruction and continuing through World War II and the civil rights movement, there arose a new set of religious organizations, denominations, and independent congregations to which Black people referred collectively as 'the Sanctified Church.' Largely associated with the Holiness movement of the late nineteenth and early twentieth century, these churches as they emerged attracted significant numbers of Black women. Drawn by the preaching of charismatic Black women and men, Black women and men abandoned Baptist and Methodist churches, which, in their zeal to regularize their worship and to assimilate culturally, had begun to trample upon the well-defended traditions of slave religion with its oral music tradition and it ecstatic praise traditions" (Gilkes 1985, p. 77).

2　The 1949 and 1951 editions of *The Negro Motorist Green Book: An International Travel Guide* (also known as *The Negro Travelers Green Book*) do not list a "Casbah Hotel" in the District of Columbia (Schomburg Center for Research in Black Culture, Manuscripts, Archives and Rare Books Division, The New York Public Library 1949, p. 15; 1951, p. 21). The popular guidebook listed "Casbah", located at 1211 U St., N. W., under "Restaurants". Kimberly Prothro Williams and EHT Traceries list Casbah as a "popular club" in their brochure about the U Street District (Williams and Traceries 2003, p. 10). Beginning in 1952, Casbah is no longer listed in the guidebook. The 1957 and 1959 editions of the *Green Book* (which coincide with Caesar's time in Washington, DC) do not list Casbah at all. Caesar mentions eating in a restaurant before going to bed (Schomburg Center for Research in Black Culture, Manuscripts, Archives and Rare Books Division, The New York Public Library 1952, 1957, 1959; Caesar 1998, p. 67). It may be that she confused the name of the restaurant.

3　Evelyn Brooks Higginbotham in Righteous Discontent notes that the "politics of respectability" proffered by African American women in the National Baptist Convention provided a "discursive common ground in its concern for sexual purity" (Higginbotham 1993, p. 198). Anthea D. Butler in Women in the Church of God in Christ notes that those "who were unmarried, male or female, were expected to be celibate". The Purity Class, established by the Women's Department of the COGIC in 1926, was the "primary vehicle for teaching sanctification in sexual matters" (Butler 2007, p. 86).

4　One exception is the Church of God in Christ denomination. A Purity Department offers classes for girls and boys, taught separately. Black gospelwoman CeCe Winans in her autobiography notes the subtle differences in the treatment of girls and boys when she participated in purity classes as a youth (Winans and Weems 1999, p. 74).

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
