# Peer review of "Shirley Caesar and the Politics of Validating Sexual Agency"

_religions, doi:10.3390/rel13060568_

Round 1
Reviewer 1 Report
I found your essay to be well written, carefully researched and compelling. I learned a good deal from reading it. Thank you.
Author Response
Thank you Reviewer 1!
Reviewer 2 Report
This is an important contribution to understanding discourses of sexuality and purity in the Black US church context. It should be published. I was glad to see all the important interlocutors here: Collins, Douglas, Moultrie, Lomax. The archival research using the “Green Book” is an interesting addition which allows this essay to speak across disciplinary specialities—the author may wish to highlight that in the article’s text, rather than burying it in a footnote. I want to make a few suggestions that will strengthen the essay’s contribution.
The primary way I think the essay can be strengthened is by clarifying the argument’s structure and some terms used. In particular, Caesar’s encounter with the singer who did not pay her fairly is fascinating (not least for the story of the pear theft which makes us think of Augustine’s in Confessions, but unlike his sin, an act of retributive justice!) However, this story which focuses on economic exploitation is not clearly stitched into the overall narrative about sexual purity. I think the common thread is exploitation by the expectations of female behavior in a male-dominated context, but if so, that should be spelled out. It’s interesting that both stories (the pay and the hotel room “attack”) are instances where Caesar ultimately triumphs. I’m sure the story belongs in the essay, but the author hasn’t quite established why it connects to what is framed as a narrative about sexual purity.
There’s also an inconsistency in the descriptions of the hotel room visit, which the author in a few places describes as attempted “holy sexual consensual seduction” (e.g. line 114). At least three of these words need to be further unpacked. Do we know this man’s intentions were consensual seduction rather than violent assault? Caesar seems to be interpreting his intentions as at least somewhat threatening due to her use of the chain on the door. And what would make this encounter, whether consensual or forced, holy?
I see that “seduce” is Caesar’s word (line 141) but given the different perspectives of earlier generations compared to now, it might be possible to name that encounters that might once have be perceived by both parties as “seduction” would now be seen as coercive. In line 343, the author describes this instance as an “attack” in quotes, but at that point it is not clear if that is Caesar’s language. The author does give the citation later where Caesar describes it as an attack. While I recognize that consensual seduction could still be viewed as a spiritual attack, I would like to see the author unpack further the contrast between Caesar’s language of seduction (consensual), attack (violent), and the fact that in the contemporary context we recognize that coercion can be present when someone consents to sex they do not want, particularly due to power dynamics around gender and religious authority.
As a minor note, I would recommend defining Feagin’s term of “open person” in line 43.
Fascinating stuff! Thank you for this essay.
Reviewer 3 Report
Summary
The author's thesis is: "Caesar's identities (Black, woman, Christian, chaste) intersecting with potential sexual coercion and her reaction (especially her insistence on maintaining her sexual purity throughout the potential assault) is fodder for the reinforcement of black male authority, power, privilege and domination in the Black Church."
The author draws on selected stories in Caesar's autobiography to support his/her argument, particularly significant is Caesar's narration of the Cabash Hotel incident during which a fellow black gospel male singer came to her hotel room to make a coded sexual invitation to her. He asked her if she would permit him to come into her hotel room so that she could "bless" his cross. The author refers to his pass at her as an attempt at "holy sexual consensual seduction." Caesar took pride in maintaining her sexual purity and saying no to the black male gospel singer's ploy to come into her room. Although she was young, she was not "naive" regarding what the gospel singer really wanted. Caesar expressed pride in resisting temptation. She did not want to be marked or judged as sexually promiscuous. The author argues that Ceasar tells this story "to state and affirm her allegiance and accountability to her faith community...to state and affirm her allegiance and accountability to God...and to affirm her submission and accountability to the Black religious male heteropatriarchal ruling authority." It is the third point that is most problematic for the author. Caesar tells the story to reinforce her purity with a "heteropatriarchy-dominated hierarchy."
The paper is well-researched and demonstrates engagement of primary and secondary sources, including leading feminist scholars who examine these subjects.
Reflection and Comments
Besides being seduced, the author acknowledges that Caesar feared being raped. As the author has currently summarized the story, I wonder if the Cabash Hotel incident is the best example to support the author's thesis.
Was the man flirting with her and attempting to engage in consensual relations? Was he a potential rapist who took advantage of young women? Does Casar's autobiography say more about the young male gospel singer? Did they have a conversation before he visited her room? Was she attracted to him? think that saying more about the young man and Caesar's potential attraction to him might help make your argument stronger regarding "consensual seduction" and Ceasar's aligning herself with normative heterosexual patriarchal ethics. You discuss how Ceasar expressed pride in resisting temptation. However, did she reflect on not becoming a potential victim of sexual assault? Does her pride in resisting temptation suggest that she was tempted or attracted to the young man and did not perceive him as a rapist? I think you are making this argument. Develop it more. Did the young male singer show up at her door late that evening because he perceived that she was attracted to him? Did Ceasar flirt with him at some point? As it stands, it seems that he randomly showed up at her door.
The potential ethical concern that I have with the paper concerns the issue of rape. Could Caesar saying no to the singer's request be grounds for a feminist reading of Caesar? It seems that saying yes to the invitation would have opened herself up to the possibility of being taken advantage of or sexually assaulted. Wasn't it wise for her to say no in this particular case and maintain her purity from sexual violence? Your thesis includes the wording "(especially her insistence on maintaining her sexual purity throughout the potential assault)." It is important to distinguish potential assault from a sexual advance. It seems right to maintain purity in the case of a potential sexual assault. Choosing to accept or not accept a consensual sexual advance is a different concern. Perhaps you might want to revisit the wording of this portion of your thesis.
Lastly, consider making a stronger connection between the story of Reverend Jones' cheating her out of money and your central argument about sexual coercion and sexual purity. Is the purpose of this story merely to demonstrate Caesar's wits and intelligence as a young gospel singer?
Round 2
Reviewer 3 Report
The author has done a great job at revising the essay and responded to suggested revisions. I think the essay is sharper and more focused. For example, I appreciate how the author has placed Caesar within the Black sanctified church rather in particular rather than the Black Church at large.
I think a few lines defining and explaining what the Sanctified Church is would be helpful. What is distinctive about this tradition as opposed to Baptists, Methodists, or the Black Church? Also, you might provide a succinct definition of sexual agency.
Perhaps lines 234-249 should be block quoted to stand out from the rest of the text. This may be something the editor can address.
Author Response
Point 1: I think a few lines defining and explaining what the Sanctified Church is would be helpful. What is distinctive about this tradition as opposed to Baptists, Methodists, or the Black Church?
Response 1: I have inserted a definition of the Black Sanctified Church in lines 77-84.
Point 2: Also, you might provide a succinct definition of sexual agency.
Response 2: I have inserted a definition of sexual agency in lines 61-63.
Point 3: Perhaps lines 234-249 should be block quoted to stand out from the rest of the text. This may be something the editor can address.
Response 3: I have indented block quote about Reverend Jones in two places: lines 244-260 and lines 277-284.
Thank YOU!